# Towards Thermally Reversible Networks Based on Furan-Functionalization of Jatropha Oil

**DOI:** 10.3390/molecules25163641

**Published:** 2020-08-10

**Authors:** Frita Yuliati, Peter J. Deuss, Hero J. Heeres, Francesco Picchioni

**Affiliations:** 1Laboratory for Polymer Technology, Agency for the Assessment and Application of Technology, Jalan M.H. Thamrin no. 8, Jakarta 10340, Indonesia; frita.yuliati@bppt.go.id; 2Department of Chemical Engineering (ENTEG), University of Groningen, Nijenborgh 4, 9747 AG Groningen, The Netherlands; p.j.deuss@rug.nl (P.J.D.); h.j.heeres@rug.nl (H.J.H.)

**Keywords:** jatropha oil, furfurylamine, Diels-Alder, experimental design, bismaleimide

## Abstract

A novel biobased monomer for the preparation of thermally reversible networks based on the Diels-Alder reaction was synthesized from jatropha oil. The oil was epoxidized and subsequently reacted with furfurylamine to attach furan groups via an epoxide ring opening reaction. However, furfurylamine also reacted with the ester groups of the triglycerides via aminolysis, thus resulting in short-chain molecules that ultimately yielded brittle thermally reversible polymers upon cross-linking via a Diels-Alder reaction. A full-factorial experimental design was used in finding the optimum conditions to minimize ester aminolysis and to maximize the epoxide ring opening reaction as well as the number of furans attached to the modified oil. The optimum conditions were determined experimentally and were found to be 80 °C, 24 h, 1:1 molar ratio, with 50 mol % of LiBr with respect to the modified oil, resulting in 35% of ester conversion, 99% of epoxide conversion, and an average of 1.32 furans/triglyceride. Ultimately, further optimization by a statistical approach led to an average of 2.19 furans per triglyceride, which eventually yielded a flexible network upon cross-linking via a Diels-Alder reaction instead of the brittle one obtained when the furan-functionalization reaction was not optimized.

## 1. Introduction

Thermosetting polymers are widely used in different applications as a consequence of their superior mechanical, chemical, and thermal properties. However, this kind of material is susceptible to irreversible damage resulting from crack formation and growth [1,2,3]. In self-healing polymers, the micro-cracks inside their structures can be easily repaired to prevent the development of larger damages, thereby improving safety, extending service life, and reducing costs [4,5].

In general, two methods have been developed for obtaining self-healing polymers; 1) incorporating healing materials into the polymers in microcapsules or vascular networks, and 2) employing mendable chemical bonds in the polymeric chemical structure. According to the first method, the healing agent containers are ruptured when the cracks develop. The healing agents bleed and polymerize to close the gaps in the material and restore the mechanical properties. This method needs no external action, but can only repair the damages once [6,7,8]. The second method needs an external stimulus, but it is regarded as superior because the healing process does not need additional components and can be performed repeatedly in the entire part of the polymers [9,10,11]. The Diels-Alder reaction (DA) is a popular method employed in self-healing polymers because of its thermally reversible nature. The DA reaction takes place at relatively low temperatures and can be reversed upon heating. Introducing DA moieties into a polymer structure enables healing or even recycling, because the corresponding adducts can be broken down into their original DA pairs upon heating [12] and formed again upon cooling [3]. In addition, the DA reaction is also considered to be efficient, versatile, and selective [13].

The most studied DA pair for thermally reversible networks is the combination of furans and maleimides [14]. This pair can be employed in modifying existing thermosetting polymers, for example, by incorporating furan groups to an epoxy resin and adding polyfunctional maleimides for cross-linking [2,5,15]. Another method is by cross-linking an epoxy resin with a diamine cross-linker having DA adducts on its structure [16,17]. New polymers can also be developed by cross-linking new monomers bearing multiple furans and maleimides [3,9,18].

The synthesis of novel thermally reversible polymers from renewable precursors is an attractive field of study, since they answer the challenge of the use of renewable feeds instead of fossil feedstocks [19,20,21]. Vegetable oils are the most important renewable feedstocks for the chemical industry [20,21,22]. Several studies have reported the use of oils from the seeds of castor, tung, soybean, and linseed in developing polymers based on the DA reaction [23,24,25,26]. Tung oil was reported to directly undergo DA reactions because two of the three conjugated C-C double bonds present in the fatty acid tails can act as a diene [23,24]. Upon mixing with various bismaleimides as dienophiles, the DA reaction occurs, resulting in a powder or gel. However, the products did not show thermoreversible properties due to extreme changes of the structure related to the formation of the adducts [23]. Other oils cannot readily undergo DA reactions and need to be modified prior to polymerization. A common modification strategy for these oils is by attaching furan groups to the oil structure, after which they are cross-linked with bismaleimides [25,26]. Furfurylamine is frequently used as the furan source. Since furfurylamine is also biobased [27], this approach ensures the complete “green” character of the polymeric material.

Furan-derivatization of acrylated soybean oil successfully yielded a monomer containing three furans in a triglyceride molecule [26]. The synthesis route consisted of three steps: epoxidation, acrylation, and subsequent furan-functionalization of the oil [26,28]. The route to attach furan units can be shortened by directly reacting epoxidized oil with furfurylamine. However, the amine in furfurylamine reacts with both the epoxide rings and the triglyceride esters (Figure 1). In a reaction between epoxidized linseed oil and furfurylamine, it was found that when all triglyceride ester groups were fully converted, not all epoxides had reacted, probably due to steric hindrance associated with two neighboring rings [29]. Furan-functionalized linseed oil was reacted with bismaleimide via a DA reaction, resulting in a polymer, which was fully soluble in 1,1,2,2-tetrachloroethane, indicating that the polymer has a linear instead of cross-linked structure [29]. This phenomena has also been reported by other authors: aminolysis of glyceride ester takes place alongside the expected oxirane ring opening during reactions between epoxidized vegetable oils and amines [30,31,32]. A general procedure for the preparation of self-healing polymers based on thermally reversible networks comprises in general two steps: the first is the functionalization of the base material (vegetable oil in the present case) with the required functional groups followed by the addition of the crosslinker (curitng step). The present works focuses on the first step. In general, the formation of strong polymeric networks requires the use of highly functionalized monomers to generate a high crosslinking density [33]. In the case of furan functionalization of triglycerides with furfurylamine, the esters bond cleavage by aminolysis should be minimized to keep the triglyceride structure intact, and therefore obtain an overall higher number of furans per monomer molecule. The use of catalysts can be advantageous in the synthesis of such molecules to steer the reactivity towards the desired pathway. Several Lewis acid catalysts such as bismuth trichloride, zinc(II) chloride, copper(II) tetraflouoroborate, and lithium bromide have been reported to successfully enhance the reaction of amines with several different epoxy compounds [34,35,36,37].

Recently, lithium bromide was employed to catalyze the reaction between epoxidized jatropha oil with furfurylamine, resulting in 47% conversion of the ester groups and 74% conversion of the epoxides. After the cross-linking with bismaleimide, a brittle polymer was obtained [38]. The reported furan-functionalization of epoxidized oil was only performed at a single reaction condition. Optimization of this reaction is desired in order to to obtain products with enhanced properties.

In this work, experimental studies are reported aimed at optimizing conditions to maximize oxirane ring opening and the number of furans attached, and also to minimize ester aminolysis. Oil from Jatropha curcas seeds was used, because of its chemical structure and anticipated future production capacity [39]. The oil was epoxidized accordingly and subsequently reacted with furfurylamine under different conditions (Figure 1i). The effects of type of catalysts, catalyst loading, temperature, reaction time, and molar ratio between the epoxides present in the oil and furfurylamine on yields were examined. The experiments were designed and statistically analyzed according to a full factorial experimental design, in order to understand how each variable and their combinations affect the reaction. This information was used to determine the optimum reaction conditions to obtain a monomer containing more than two furan units in each oil-derived molecule. Subsequently, cross-linking via a Diels Alder reaction (Figure 1ii) was performed and relevant product properties were determined.

## 2. Results and Discussion

### 2.1. Epoxidation of Jatropha Oil

The composition of jatropha oil was analyzed by ^1^H NMR to determine the average number of unsaturations in a triglyceride molecule. This information is required to determine the stoichiometry of reactants for the epoxidation reaction. For this purpose, the ^1^H NMR signals at δ 2.75 and 2.01 ppm assigned to allylic protons were used with the terminal –CH_3_ signal at δ 0.88 ppm as the internal standard (Figure 2). The measurement revealed that the jatropha oil used in this study had an average of 3.62 C-C double bonds per triglyceride molecule.

Epoxidation was perfomed by reacting the oil with performic acid synthesized in situ. The extent of the epoxidation reaction was also evaluated by using ^1^H NMR analysis (Figure 2), employing the new signals that appeared between δ 2.82–3.18 ppm arising from the epoxide units. The epoxidized oil was found to contain between 3.15 and 3.74 oxirane rings per triglyceride upon application of standard epoxidation conditions. This variation between different batches is likely the result of the inhomogeneous composition of the oil, different degrees of epoxidation, as well as some experimental error related to the NMR measurement. The following furan-functionalization reaction was performed based on the measured average epoxide number per molecule of each batch. Moerover, the number of incorporated furans (see below) is always normalized to the number of ester and epoxide groups reacted, thus factually corrcecting for small differences between batches.

### 2.2. Furan-Functionalization of the Epoxidized Jatropha Oil

The next step involved attaching furan units onto the epoxidized oil structure. Furfurylamine was used as a potential furan source for DA reactions, with the amine groups expected to react with the epoxide groups of the epoxidized jatropha oil. A full factorial experimental design was used to quantify the effects of process variables on reaction efficiency. Three variables were employed, namely; temperature, time, and the molar ratio between epoxides and furfurylamine, according to Table 1. We started with a set of reactions without catalyst, which later would be used as a base condition to evaluate the performance of the catalysts. Similar conditions were applied in reactions using LiBr at two different concentrations, determined as the molar percentage of LiBr with respect to the amount of epoxidized oil, with the assumed structure given in Figure 1. The upper levels of reaction temperature, time, molar ratio, and LiBr loading were selected based on a previous work [39]. The lower levels of temperature and molar ratio were selected to understand the minimum extent of reaction (i.e., under experimental conditions, which do not favor the desired reaction). The lower level of LiBr loading was determined according to a report on oxirane ring opening using LiBr as catalyst [37].

Our experiments included reactions with an excess of furfurylamine, which had to be removed from the product mixtures prior to analysis to avoid further reaction. Since furfurylamine is highly soluble in water, liquid-liquid extraction of furfurylamine with water was preferred over evaporation. This aqueous extraction should also remove the added catalyst. The reaction product was dissolved in chloroform to prepare it for excess furfurylamine removal and cross-linking with bismaleimides according to reported methods [38,40,41]. The remaining furfurylamine was extracted with water. A single extraction step proved to be sufficient to remove all furfurylamine from the products, as shown in the ^1^H NMR spectra taken afterwards (Appendix A). The signal at δ 3.80 ppm, which is specific for furfurylamine (protons at the α-position to the primary amine), disappeared from the spectra after the extraction.

After furfurylamine and solvent removal, the products were analyzed using ^1^H NMR. The spectra differ considerably from that of the epoxidized oil (Figure 3). The signals between δ 2.82–3.18 ppm (c) representing epoxides gradually decreased with more severe reaction conditions. The area of the triplet at δ 2.35 ppm (b) assigned to a proton at the α position with respect to the carbonyl group decreased, while the area of a new signal at δ 2.19 ppm (h) showed an increase. This new signal was assigned to the -CH_2_ located next to an amide resulting from the reaction between an ester and furfurylamine. The furan groups attached to the triglycerides were indicated by signals at δ 7.35 ppm and two signals between δ 6.10–6.43 ppm, all showed increasing areas under more severe reaction conditions. The signals b, c, e, f, g, and h were used to further evaluate the outcome of the reactions.

As illustrated in Figure 1 (top), the reaction of the amine can take place at the epoxy and ester groups. The corresponding conversion values together with the total incorporated furan groups are plotted in Figure 4 to provide an insight into extending the relative selectivity of the furan incorporation. For the reactions without catalyst, it is clear that the esters reacted more easily with furfurylamine compared to the epoxides, as also observed in previous studies [29,32,38]. Of all batches performed, the average ester conversion (46%) was higher than that of the epoxides (36%). The result is in agreement to a recent report stating that internal epoxides are not very reactive towards amines [30]. However, it was found in our experiments that both at 80 and 30 °C, epoxide conversion can be similar to ester conversion, which occurred when the reaction proceeded in 5 h (Figure 4). Afterwards, the esters continued to react while the epoxide conversion was not increased. Some fatty acid chains in the epoxidized jatropha oil contain two epoxy rings. One of these neighboring groups can be less reactive because of steric hindrance [29], leading to lower epoxide conversion upon prolongation of the reaction time.

As observed in Figure 4, the average number of attached furans per triglyceride structure is always lower than the number of esters and epoxides converted, indicating that side reactions occurred (Appendix A). The furan signals on the ^1^H NMR spectra do not show on which site (ester or epoxide) they are attached (although this could be seen on ^13^C-NMR spectra), therefore we relied on the ester and epoxide conversion data to predict the structure of the product. The effect of time on the conversion values is generally quite clear as longer times (24 vs. 5 hrs) result in higher number of furans attached as well as of ester conversion. However, for the reaction at 80 °C at equimolar ratios (80,5, 1 vs. 80, 24,1), a significant decrease in the ester and epoxy conversion in time is observed. The reasons for this behavior are yet to be studied in detail, possibly in combination with an in depth kinetic studies of the competing reactions. Additionally, an increase in the molar ratio between amine and epoxides results generally in higher conversion values (for both epoxy and ester groups), while the number of attached furans remains constant. As a consequence of our definition of attached furans (normalized for the number of epoxy and esters groups converted), this implies that the occurrence of side reactions becomes more relavant when using an excess of furfurylamine. By following a similar reasoning, one might observe that also higher temperature seems to favor side reactions. Although the nature of these side recations is not clear at the moment, one might easily speculate that they probably involve the furan moiety based on their relatively low stability in a variety of reactions [42]. Additionally, the higher ratio of furan combined with a higher temperature can change solvation behaviour and therefore reactivity of the different components. In addition to all the effects highlighted above, it is worth noticing how the highest furan functionalization level (27%) was achieved in a reaction at 80 °C, 24 h, and a 1:5 molar ratio of epoxide to furfurylamine, abbreviated as reaction {80,24,5}. However, 71% of the esters were converted, suggesting that most of the triglyceride structure was lost to mono- and di-glycerides as well as free fatty acids chains. A more promising result was obtained by a reaction at 80 °C, for 24 h, and using a 1:1 molar ratio of epoxide to furfurylamine (reaction {80,24,1}). Although the number of attached furans was lower (24%), the ester conversion was also low (33%), which indicates that most of the triglyceride structure was still intact.

In a previous work [38], furan-functionalization of epoxidized jatropha oil with 50% of LiBr loading yielded 74% of epoxide conversion and 47% of ester conversion. We intended to improve this result by varying the reaction conditions as described in Table 1. The reaction products were analyzed by using the same method as the reactions without catalyst (Figure 5).

A preliminary study (not shown for brevity) showed that the use of 50% of LiBr improved epoxide conversion during the reaction with furfurylamine. LiBr was found to be an efficient catalyst for epoxide ring-opening reaction by amines, as a consequence of the Lewis acid character of the lithium ion that activated oxygen-containing electrophiles for nucleophilic attack [37]. The use of such a high concentration of LiBr was shown to be necessary in a previous work [38]. Besides being active as a catalyst, we cannot exclude a priori that other effects (for example, increased medium polarity by addition of LiBr) plays an important effect.

Although the epoxide conversion was improved, the ester remains typically more reactive than the epoxides as depicted in Figure 5. Moreover, despite the increased epoxy conversion, the number of attached furans remains fairly constant. This implies once more that side reactions are taking place. As also observed in previous sets of experiments, the reaction performed at 80 °C for 24 h with equimolar epoxide and furfurylamine amounts (reaction {80,24,1}) gave a high epoxide conversion, while the ester conversion remained low. Thus we can conclude that the use of LiBr is promising to obtain a monomer suitable for building thermally reversible networks, because a large portion of the esters remained in the product and a higher furan functionality in a molecule is demonstrated. The resulting product has 99% of epoxide conversion, 35% of ester conversion, and on average 1.32 furans attached to each triglyceride molecule or 28% of the total converted epoxide and esters.

In order to get a deeper understanding on the effect of reaction conditions on the reaction, the sets of reactions without catalyst and with LiBr were evaluated using a statistical approach. The statistical analysis provides information of which variables or factors have significant influences on the reaction, as indicated by the measured responses, i.e., ester conversion, epoxide conversion, and furans attached. The significance of the variables on the measured responses are shown by the Pareto graphs in Appendix A. The x-axis of the diagram represents the t-value of the data. When the t-value exceeds a critical value, which is indicated by the red line, the reaction variable has a significant effect on conversion [43].

Appendix A shows that ester conversion was mostly influenced by the molar ratio of furfurylamine to epoxides. This tendency explained why the reaction {80,24,5} in Figure 5 results in higher ester conversions than the reaction with lower molar equivalents {80,24,1}. Ester aminolysis already takes place under mild conditions (see above), therefore a higher amount of furfurylamine results in higher rates for the reaction between the amines and esters to occur.

The effect of reaction conditions on epoxide conversion is more complex than that for ester conversion, as given in Appendix A. Temperature has the strongest influence on epoxide conversion. Applying a higher temperature decreases the viscosity of the oil [44] and improves its mobility. Higher oil mobility can reduce the negative effect of steric hindrance, thus avoiding the higher selectivity of the reaction towards ester aminolysis [29,38]. On the other hand, it is also known that the reactivity of the epoxy groups towards nucleophilic substitutions increases with temperature [45,46], thus the observed trend in the data can be ascribed to a kinetic effect. Other significant factors influencing the epoxide conversion are the reaction time and the catalyst loading. The combination of time and temperature also significantly affect the conversion. On the other hand, the furfurylamine concentration has a minor effect on the epoxide conversion.

Beside ester and epoxide conversion, the amount of furan incorporation has to be taken into consideration. The average number of furans is always lower than the total esters and epoxides converted (Figure 4 and Figure 5), indicating the occurrence of side reactions (as described in the Appendix A). However, the statistical analysis reveals the factors that influence the amount of furans attached (Appendix A). According to the Pareto diagrams in Appendix A, furan attachment was mostly influenced by the molar ratio of furfurylamine to epoxide. Reaction time and temperature have an effect as well but to a smaller extent, followed by the combination of several variables.

According to the results showed in the Pareto graphs, surface plots were constructed by using the most significant factors as the x and y axes, with the responses given on the z axis (Figure 6). These plots clearly show how the changes in the factors affect the reaction products.

From the trends in Figure 6, we can conclude that ester conversion can be minimized and epoxide conversion maximized when using a low furfurylamine concentration in combination with a long reaction time and a high LiBr loading. Since temperature influences both the conversion of the epoxides and the esters, a moderate temperature should be applied to increase the epoxide conversion whilst keeping the ester conversion at an acceptable level.

Since furan attachment and ester conversion were mostly prompted by the same variable, they showed similar trends, as shown in Figure 6. Although we would like to have opposing results between the esters conversion and furans attached, there were opportunities to improve the level of furans attachment while increasing the selectivity towards epoxide conversion. Increasing the reaction time and temperature was a promising direction to obtain a more desired product mixture.

A number of experiments were performed to optimize the reaction. Figure 7 displays the results of these experiments {80,24,1,50} with 50% of LiBr as a base condition. The results show that longer reaction time increases furan attachment and decreases epoxide conversion. When the LiBr loading was increased to 150%, similar conversions of esters and epoxides were obtained. The use of 100 mol % of LiBr combined with increasing the temperature to 100 °C retained more esters than 150 mol % and generated a higher number of furans attached to the oil structure. Therefore, the condition of {100,24,1,100} was selected to yield furan-functionalized jatropha oil for the synthesis of polymers in our further study. In this product, 86% of epoxides and only 4% of esters were converted, yielding an average of 2.19 furans on each triglyceride molecules.

### 2.3. Polymer Synthesis Via a Diels-Alder Reaction

The product mixtures of {100,24,1,100} and {80,24,5,50} were each reacted with an aliphatic bismaleimide. The product mixture {100,24,1,100} is expected to retain most of its triglyceride structure, because only 4% of the esters were converted. On the other hand, the product {80,24,5,50} has an ester conversion of 70%, epoxide conversion of 95%, and an average of 3.75 furans attached to each triglyceride molecules. The obtained polymers show different flexibilities and {100,24,1,100} produced more flexible polymer (Figure 8). A polymer made from this batch is expected to be flexible, owing to the long chains of the modified oil. The number of furans attached to this structure also implies that reaction of the monomer with a bismaleimide would yield a polymer with a low cross-link density. The batch {80,24,5,50} with high ester conversion can have a higher number of furans attached, but the structure will be more likely to consist of free fatty acid amides with one to three furans units in each molecule. This monomer mixture is expected to yield more brittle polymers than {100,24,1,100} because of the shorter chain length and higher cross-link density.

Moreover, solvent test revealed that almost 20% of the polymer from {80,24,5,50} became insoluble gel, while the entire sample from {100,24,1,100} was completely soluble. Therefore, it was found that the brittleness of the polymer made of the shorter monomer chain mixture not only arises from the molecular size, but also from the cross-links

## 3. Materials and Methods

### 3.1. Materials

Jatropha curcas oil was obtained from Dilligent Energy System, Eersel, The Netherlands. Hydrogen peroxide solution (30%), formic acid (≥98%), sodium chloride (≥99%), and furfurylamine (≥99%), were purchased from Sigma Aldrich, Munich, Germany; peroxide test strips (0.5–2–5–10–25 mg/L H₂O₂) was obtained from Merck, Darmstadt, Germany; toluene (≥99.5%) was purchased from Macron Fine Chemicals, Deventer, The Netherlands; whereas lithium bromide (≥98%) and sodium chloride (≥99%) were obtained from Fluka, Landsmeer, The Netherlands; 1,12-bis(maleimide)dodecane was prepared according to a previous work [47].

### 3.2. Methods

#### 3.2.1. Epoxidation of Jatropha Oil

Epoxidation of jatropha oil was conducted according to a method used in a previous work in our group [38] with some modification. Here, 20 mL (0.02 mol) of jatropha oil was dissolved in 100 mL of toluene and heated to 70 °C while being stirred at 700 rpm with a magnetic stirrer, in a three-necked round-bottomed flask equipped with a condenser and two stoppers. To this flask, 150 mL of H_2_O_2_ solution (1.47 mol H_2_O_2_) and 12 mL (0.53 mol) of formic acid were added drop wise separately during the first hour of stirring. The mixture was continuously stirred and heated for 24 h. The oil phase was separated from the water phase by using a separation funnel. The oil phase was then washed with 5% of sodium chloride solution in water until no H_2_O_2_ was detected by using peroxide test strips. Subsequently, toluene was evaporated to obtain the epoxidized jatropha oil used for reactions with furfurylamine. Among different batches of epoxidized oil produced, only the ones with full conversion of the C-C double bonds were used in the furan-functionalization reactions.

#### 3.2.2. Furan-Functionalization of the Epoxidized Jatropha Oil

Epoxidized jatropha oil (0.5 g, 0.0005 mol), furfurylamine, and catalyst were mixed in 4-mL vials and closed with caps. The mixture was heated to the desired temperature and stirred at 500 rpm with a magnetic stirrer. The experimental conditions were set up according to two-level full factorial experimental design with two replicates for each variation (Table 1). Two sets of experiments, viz. without catalyst and with LiBr were varied in terms of temperature, time, molar ratio, and catalyst loading for those using catalysts. Molar ratio refers to the ratio of existing oxirane rings to furfurylamine. Catalyst loading refers to the molar percentage of catalysts with respect to the amount of the epoxidized oil with full olefin conversion, assumed to have the structure given in Figure 1.

Purification of the products was performed by liquid-liquid extraction with water. The products were dissolved in 5 mL of chloroform and unreacted furfurylamine (and catalyst) was extracted with 125 mL of Mili-Q water. Chloroform was subsequently removed by evaporation.

#### 3.2.3. Characterization by Using ^1^H NMR

Jatropha oil, epoxidized oil, and the products of furan-functionalization reaction were characterized by using Varian Oxford 300 MHz NMR (Agilent, Santa Clara, CA, USA), with CDCl_3_ as a solvent. The spectra were processed by using MestRenova 12 software (Santiago de Compostela, Spain). The signal assigned to –CH_3_ at δ 0.88 ppm was used as an internal standard for all calculations.

The number of double bonds in the jatropha oil was determined by using the signal at δ 5.34 ppm, which is assigned to the vinyl protons [48,49]. The signal at δ 2.01 and 2.75 ppm were also examined, since they represent allylic and bisallylic protons in the C16:1, C18:1, and C18:2 fatty acid chains of the oil [48,49]. The extent of the epoxidation reaction was determined by the area reduction of the relevant peaks. Complete disappearance of these signals indicated that all the double bonds were converted. The number of epoxides formed in the reaction was calculated by using the signals at δ 2.82–3.18 ppm [38].

Three items were calculated as measures of the success of the reaction, namely triglyceride ester and epoxide conversions, and furans attached. In a previous work [38], ester conversion was calculated based on a peak at δ 4.44 ppm, which was considered to represent protons attached to the α-carbon with respect to the amino group, after it was reacted with triglyceride esters. Although the signal representation is correct, a partial overlap with the signals form the same moiety attached to the carbonyl (during amide formation) might still take place. Therefore, we selected a triplet at δ 2.30 ppm instead, representing a proton at the α position regarding the carboxylic ester [38]. The conversion of epoxides was calculated in similar way to the esters, on the signals related to the epoxide rings. The calculation was performed according to Equation (1).
(1)$$\mathrm{Conversion}=\frac{A_{0}{-A}_{1}}{A_{0}}$$

With A_0_ as the area of initial signals at δ 2.30 ppm and 2.82–3.18 ppm in the epoxidized oil spectra, and A_1_ as the area of the same signals in the ^1^H NMR spectra of the reaction products.

The number of furans attached to the oil structure was determined by signals at δ 7.34 ppm and between 6.18–6.40 ppm, each represents one of the protons attached to the furan rings. The calculated number of furans were compared to the maximum number of furans expected to be present based on the sum of converted esters and epoxides. We use this definition of attached furans as it immediately illustrates the occurrence of side reactions (the complementary of these values to 100% represents the percentage of epoxide and ester groups that reacted but did not result in any attachment of furan groups). This ratio was then used in the statistical analysis as a measure of the extent of reaction. The furan conversion values calculated on the basis of the maximum epoxide and ester groups conversion are provided in Appendix A.

#### 3.2.4. Statistical Analysis of the Factorial Experimental Design

Statistical analysis was performed by using the analysis of variance (ANOVA) method, using the Minitab 17 software package. The data was modeled by using a second-order polynomial regression model as given in Equation (2) [43].
(2)$$y{=\beta }_{0}+\overset{k}{\underset{j=1}{\sum }}\beta_{j}x_{j}+\sum \overset{k}{\underset{i<j}{\sum }}x_{i}x_{j}++\sum \overset{k}{\underset{j=1}{\sum }}\beta_{\mathrm{jj}}x_{j}^{2}$$

In the equation, y is the response or the dependent variables, i.e., ester and epoxide conversion, and attached furans; β_0_, β_i_, and β_ij_ are regression coefficients; while x_i_ and x_j_ are the experiment factors or independent variables, viz. temperature, molar ratio, time, catalyst, type, and catalyst loading.

Reactions between epoxidized jatropha oil and furfurylamine were designed according to Table 1. The extent of the reactions were analyzed by using the data of ester and epoxide conversion, and furans attached. These data was analyzed statistically by employing the ANOVA method [50]. The significance of each variable towards the responses is displayed in Pareto diagrams. The x-axis represented standardized effects, which was the t-values of the data. In the diagram, a reference line was drawn as a limit of whether a variable was significant. This line showed the critical value for Student’s *t*-distribution at the degree of freedom of the error with default confidence level of 0.05 [51].

#### 3.2.5. Synthesis of Thermally Reversible Polymers

Furan-functionalized jatropha oil was dissolved in chloroform with a concentration of 10 *wt*. %. 1,12-bis(maleimide)-dodecane was added with a ratio of 1:1 of maleimides to furans. The solution was stirred for 24 h at 50 °C. The solvent was evaporated, and the polymer was further annealed at 50 °C under vacuum. The occurrence of cross-linked polymers were subsequently measured by a gel content experiment according to a reported method [41] with some modification. An amount of 0.5 g of a thermally reversible polymer sample was added to 5 g of chloroform and let sit at room temperature for 3 weeks. The solution was filtered with filter paper and the gel retained by the filter was weighed.

## 4. Conclusions

Thermally reversible networks from biobased sources offer a solution for concerns around depleting petroleum resources and the difficulties to repair and recycle conventional polymeric networks. In this work, a biobased monomer for thermally reversible networks based on the Diels-Alder reaction was synthesized from jatropha oil. The oil was epoxidized to provide a reactive site at which the furans can be attached via a ring opening reaction with furfurylamine. Unfortuantely, furfurylamine also reacted with the ester groups of the triglyceride in an aminolysis reaction. This reaction lead to short monomers, which eventually generated brittle polymers.

Different reaction conditions were employed in order to maximize the number of furans attached to the oil, preferably by maximizing the epoxide ring opening reaction and minimizing the ester aminolysis. In most experiments, the ester was more easily converted than the epoxides, except in the reactions d at 80 °C for 24 h, with 1:1 molar ratio of furfurylamine to the epoxides. With the use of 50 mol % of LiBr loading at this condition, epoxide conversion was enhanced to almost 99%, while the ester conversion was suppressed to 35%. The number of furans attached was 1.32 per triglyceride, or 28% of the converted esters and epoxides. Optimization of the conditions according to a statistical model yielded an average of 2.19 furans per triglyceride, obtained for a reaction performed at 100 °C, for 24 h, a 1:1 molar ratio, and a stoichiometric loading of LiBr with respect to the epoxidized oil.

The optimized product was used in the synthesis of a thermally reversible network via a Diels-Alder reaction with 1,12-bismaleimidododecane. The generated polymer was found to be more flexible than the one synthesized from the non-optimized monomer due to the longer chain of its precursor and lower cross-link density.

This work provides the proof of principle of tailoring the properties of vegetable oil-based thermally reversible networks by optimizing the conditions for monomer synthesis. In future research, polymer properties will be tailored by selecting the structure of both the bismaleimides and the furan-derivatized oils.

## Figures and Tables

**Figure 1 molecules-25-03641-f001:**
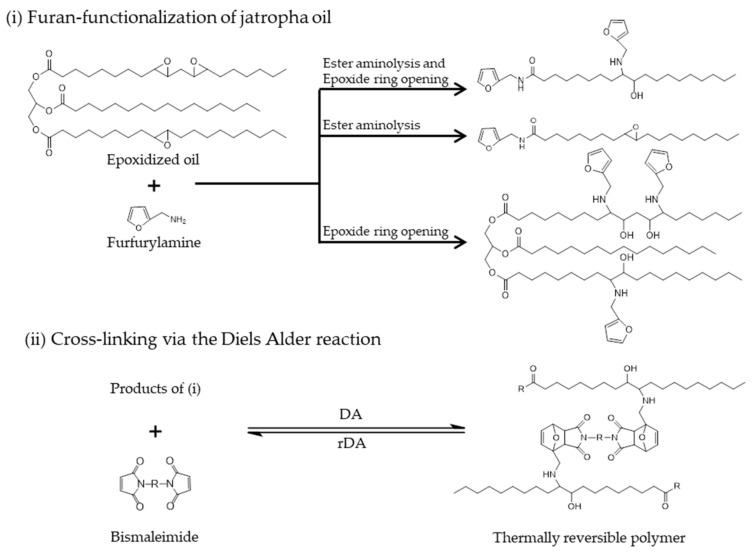
The proposed synthesis route for thermally reversible polymers from epoxidized jatropha oil.

**Figure 2 molecules-25-03641-f002:**
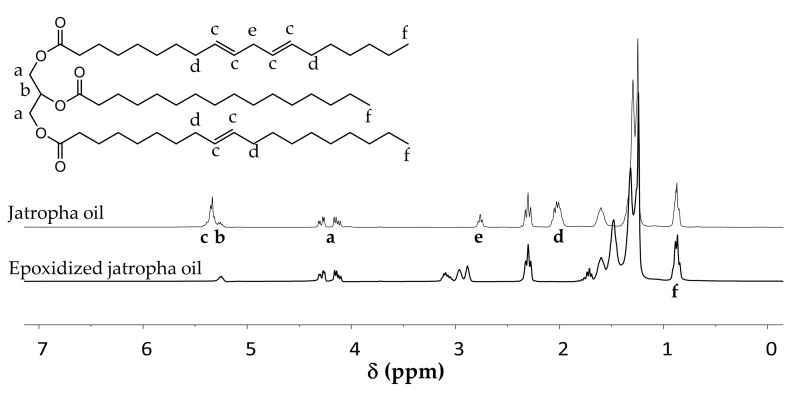
Proton NMR spectra of jatropha oil and epoxidized jatropha oil.

**Figure 3 molecules-25-03641-f003:**
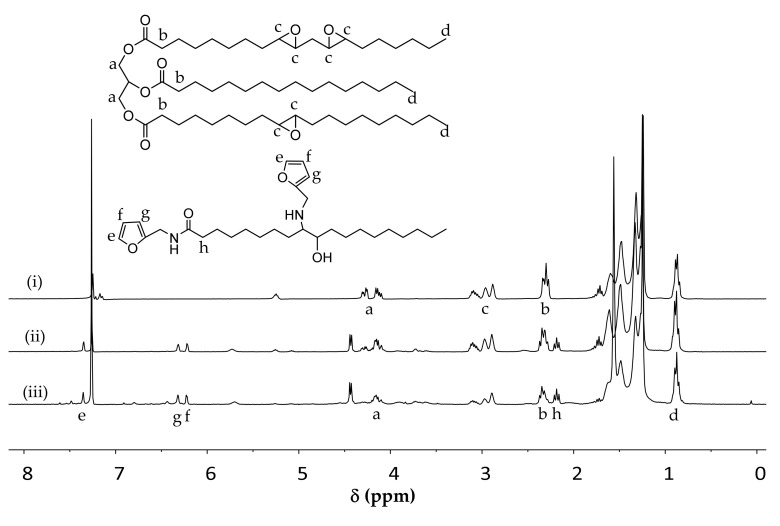
Proton NMR spectra of products of reactions without catalyst: (**i**) epoxidized oil, (**ii**) 80 °C, 1:1, 5 h, (**iii**) 80 °C, 1:5, 24 h.

**Figure 4 molecules-25-03641-f004:**
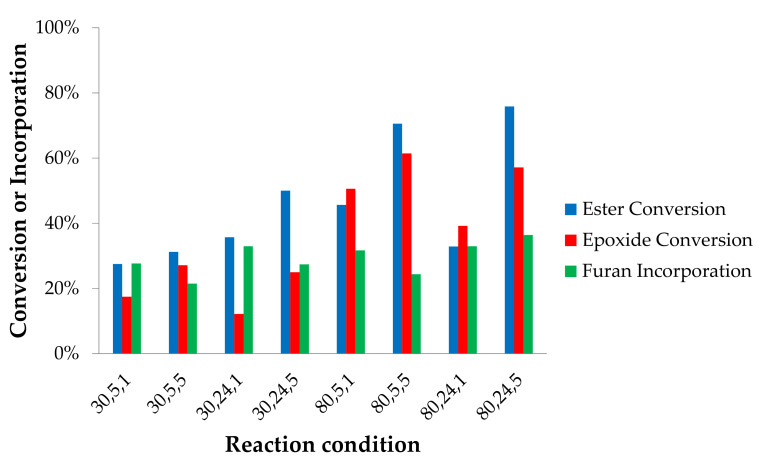
Ester and epoxide conversion, and average amount of furan attachment of the reactions without catalyst. The x-axis indicates reaction conditions: temperature (°C), time (h), and molar ratio of furfurylamine to epoxides.

**Figure 5 molecules-25-03641-f005:**
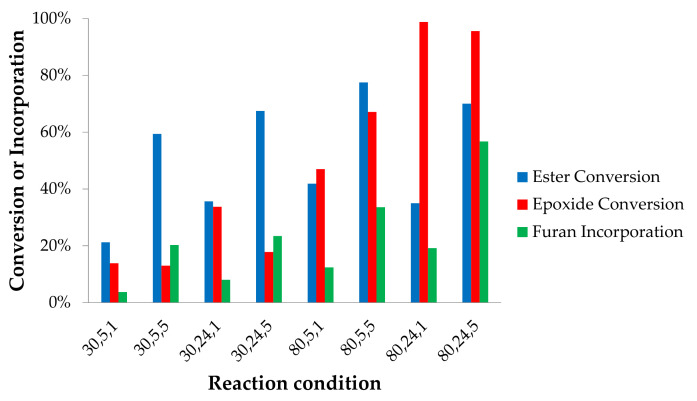
Ester and epoxide conversions, and furans attachment of the reactions with 50% of LiBr loading. The x-axis indicates reaction conditions: temperature (°C), time (h), molar ratio of furfurylamine to epoxides.

**Figure 6 molecules-25-03641-f006:**
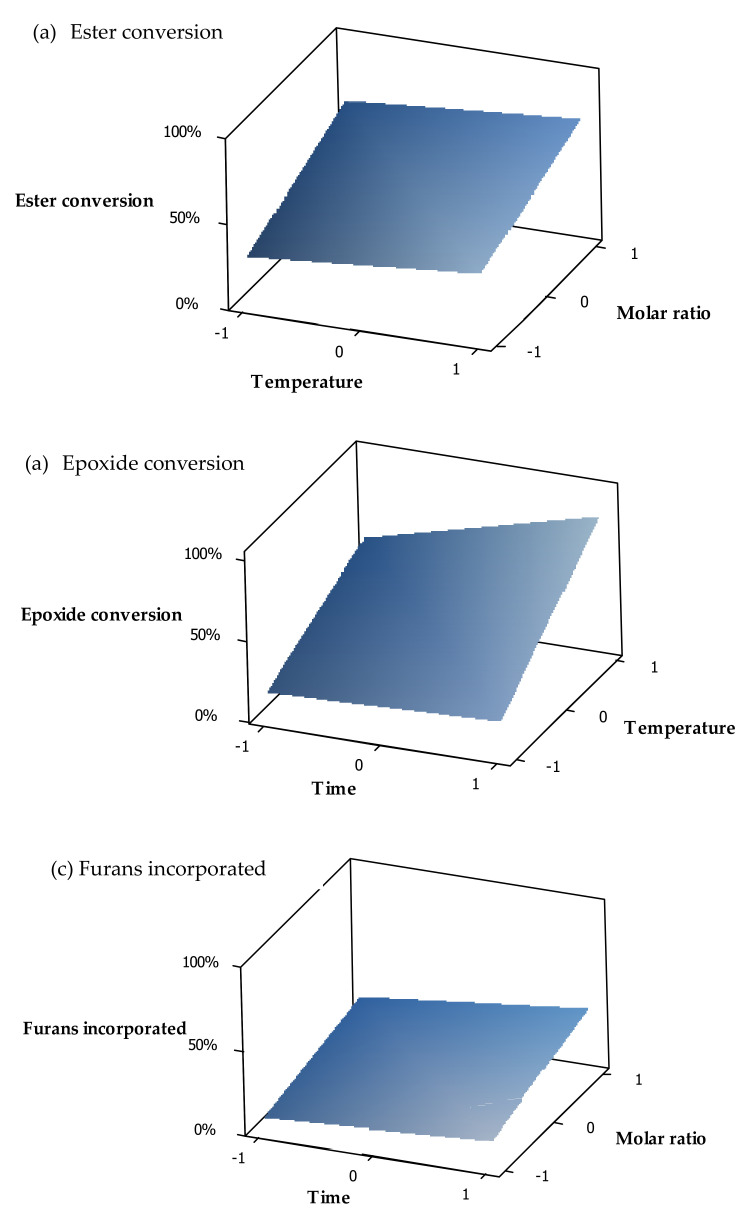
Surface plots for various responses: (**a**) ester conversion, (**b**) epoxide conversion, and (**c**) furans incorporated.

**Figure 7 molecules-25-03641-f007:**
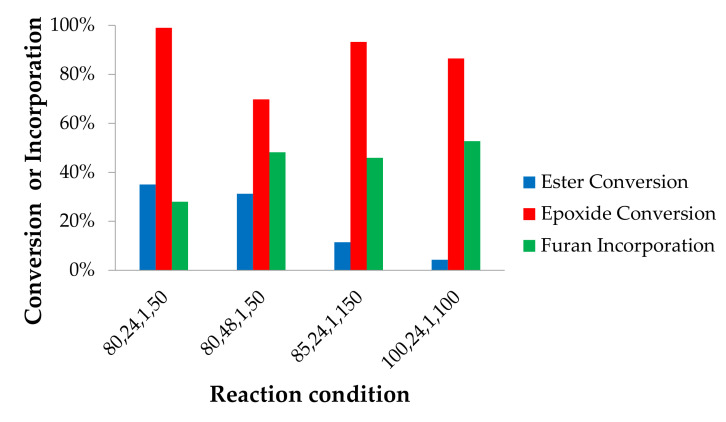
Ester and epoxide conversion and furan attachment in the presence of LiBr at various reaction conditions. The x-axis represents reaction temperature (°C), time (h), molar ratio of furfurylamine to the epoxides, and LiBr loading (% of epoxidized oil).

**Figure 8 molecules-25-03641-f008:**
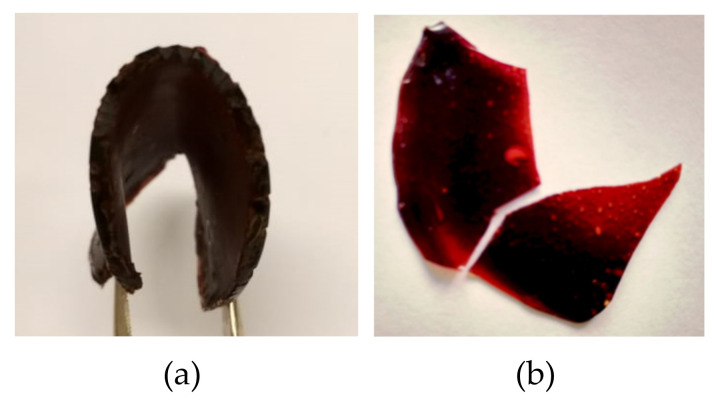
Polymer (**a**) made of monomer {100,24,1,100} was more flexible than polymer (**b**) from monomer {80,24,5,50}, the latter was broken while being bent.

**Table 1 molecules-25-03641-t001:** Overview of Experiments.

Catalysts	Variations		
	Variables	Low	High
Without catalyst	Temperature (°C)	30	80
	Time (h)	5	24
	Molar ratio	1:1	1:5
With LiBr	Temperature (°C)	30	80
	Time (h)	5	24
	Molar ratio	1:1	1:5
	LiBr loading (mol %)	5	50

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
