# Peer review of "Towards Thermally Reversible Networks Based on Furan-Functionalization of Jatropha Oil"

_molecules, 2020, doi:10.3390/molecules25163641_

Round 1
Reviewer 1 Report
The authors describe the epoxidation and subsequent furan-functionalization of jatropha oil to produce a novel monomer to create reversible polymer networks using the Diels-Alder reaction. A thorough experimental study was performed to optimize the furan-functionalization published earlier to yield the most desirable furan-functionalized monomer, finally resulting in a flexible reversible polymer network. The work and experimental procedure are very interesting, but some changes are necessary to how the results are presented.
Important issue that needs to be addressed to assess the soundness of the comparison of the results for the optimization and subsequent discussion:
Special focus should be drawn to the furan conversion as described and compared in the study. An equation to define the furan conversion should be available in the manuscript. "The calculated number of furans were compared to the number of furans expected to be present based on the sum of converted esters and epoxides." defines a value that depends on the sum of the ester and epoxide conversions, which makes comparison of different difficult. "This ratio was then used in the statistical analysis as a measure of the extent of reaction." It is not clear from this statement what exactly happens to this value numerically. Would it not make more sense to define this value with reference to the starting molecule, i.o. to the sum of converted esters and epoxides? This would result in a more absolute value, allowing more easy comparison between the different experiments. Maximum 3 epoxides and 3 esters can be converted. 100% could either be with reference to the amount of expoxide rings present in the original molecule (3), or with reference to both epoxides and esters (6 total).
Comments:
- Section 2.1: Could the average C=C value of the oil be related to the average epoxide value (and standard deviations)? An average is given for the former, while two value or a range (this is not clear) were given for the latter. This would allow to further distinguish between the variation on the C=C content of the oil and the extent of epoxidation.
- From line 167 the most important paragraphs of the manuscript start. The results are elegantly condensed. Therefore, more detail should be provided for the reader to understand the complex, yet very interesting study, which holds substantially more information than discussed. Start the first paragraph with a reference to Figure 4 while explaining the competition between ester amination and epoxide ring opening.
- More conclusions can be drawn from Figure 4 and the statistical assessment:
- Effect of time could be further elaborated. Comparison of 5 h and 24 h always results in higher furan functionalization and ester amination. For the 80 °C, 1:1 experiment the epoxy conversion drops significantly from 5 h to 24 h. Why? It would have been interesting to perform time-resolved experiments at more time instances (e.g. every 1 h), rather than just 2 times, to have more kinetic information.
- Increasing the molar ratio of amine/epoxy results in higher ester amination and epoxy conversion, while furan conversion is lower. This is definitely worth to discuss.
- At higher temperatures, the furan conversion remains almost the same, while ester and epoxide conversions increase --> less side reactions at lower temperature? Which side reactions?
- The use of the LiBr catalyst increases the epoxy conversion, yet the furan conversion is slightly lower than in the absence of the catalyst. Please elaborate what happened to the epoxide rings.
- In the paragraph starting at line 232 dealing with the effect of temperature something needs to be said about how the kinetics of both competing reactions are influenced by temperature. This effect is likely to be much greater than a decrease in viscosity of an oil. An influence of temperature and viscosity on steric hindrance would be negligible.
- In Figure 6 planes are shown, while only very few discrete data points are available from experimental results. Please correct by clearly indicating the experimental results, e.g. with a different colour or by removing the plane altogether as it has no factual significance.
- I would suggest to move the Materials and Methods section in front of the Results and Discussion section. Knowledge of the way the conversions are calculated is important to understand the results. Alternatively, the conversions should be clearly defined in the body of the Results and Discussion section.
- What happens with the catalyst at the end of the reaction?
Minor corrections:
A part of the manuscript (mainly in the introduction) is written in the “we” form. Please, convert to an impersonal and factual writing style.
On line 71 there is a broken citation reference.
On line 198 a percentage of 50% LiBr is stated. What kind of percentage is this? With reference to what? Mole percentage compared to epoxy? Also state that the loading percentages in the remainder of the manuscript are defined in the same way, or correct in the entire manuscript.
On line 266 “while increasing” i.o. ”whileincreasing”
On line 285 an “{“ is missing for 1 of the experiments
Reviewer 2 Report
The idea of this study is very useful and interesting and adds important information about the self-healing polymers via Diels–Alder reaction by using vegetable oils (jatropha oil). Different reaction conditions have been investigated for optimization of number of furan attached to the oil and the conclusions revealed that the ester was more easily converted than the epoxides and the addition of LiBr as catalyst enhanced the epoxide conversion to almost 99%.
Considering the reasonable idea and reasonable conditions investigated, I recommend the publication of this manuscript after minor revisions.
(1)The authors should correct some English errors and typos: page 2, line 71 and 77; page 4, line 148; page 5, line 180-the 13C NMR can solve the attaching to the ester group; page 9, lines 287-288-rephrase the sentence.
(2) Introduction is too long, a general scheme for obtaining self-healing polymers would be more appropriate instead the second paragraph.
Round 2
Reviewer 1 Report
The comments have been adequately revised and answered, making the discussion of the interesting results more sound.